# Night Interruption with Red and Far-Red Light Optimizes the Phytochemical Composition, Enhances Photosynthetic Efficiency, and Increases Biomass Partitioning in Italian Basil

**DOI:** 10.3390/plants13223145

**Published:** 2024-11-08

**Authors:** Soheil Fallah, Sasan Aliniaeifard, Mahboobeh Zare Mehrjerdi, Shima Mirzaei, Nazim S. Gruda

**Affiliations:** 1Photosynthesis Laboratory, Department of Horticulture, Faculty of Agricultural Technology (Aburaihan), University of Tehran, Tehran 14395-547, Iran; soheilfallah@ut.ac.ir (S.F.); mzarem@ut.ac.ir (M.Z.M.); shimamirzae24@ut.ac.ir (S.M.); 2Controlled Environment Agriculture Center (CEAC), Faculty of Agricultural Technology (Aburaihan), College of Agriculture and Natural Resources, University of Tehran, Tehran 14395-547, Iran; 3Department of Horticultural Science, INRES-Institute of Crop Science and Resource Conservation, University of Bonn, 53121 Bonn, Germany

**Keywords:** phytochemical enhancement, light spectrum manipulation, biomass allocation, *Ocimum basilicum*, controlled environment agriculture, photosynthesis

## Abstract

Controlled environment agriculture is a promising solution to address climate change and resource limitations. Light, the primary energy source driving photosynthesis and regulating plant growth, is critical in optimizing produce quality. However, the impact of specific light spectra during night interruption on improving phytochemical content and produce quality remains underexplored. This study investigated the effects of red (peak wavelength at 660 nm) and far-red night interruption (peak wavelength at 730 nm) on photosynthetic efficiency, biomass distribution, and phytochemical production in Italian basil (*Ocimum basilicum* L.). Treatments included red light, far-red light, a combination of both, and a control without night interruption. Red light significantly increased chlorophyll a by 16.8%, chlorophyll b by 20.6%, and carotenoids by 11%, improving photosynthetic efficiency and nutritional quality. Red light also elevated anthocyanin levels by 15.5%, while far-red light promoted flavonoid production by 43.56%. Although red light enhanced biomass, the primary benefit was improved leaf quality, with more biomass directed to leaves over roots. Far-red light reduced transpiration, enhancing post-harvest water retention and shelf life. These findings demonstrate that red and far-red night interruption can optimize phytochemical content, produce quality, and post-harvest durability, offering valuable insights for controlled environment agriculture. Future research should focus on refining night interruption light strategies across a broader range of crops to enhance produce quality and shelf life in controlled environment agriculture.

## 1. Introduction

The production of plants under artificial light is a modern agricultural technology that optimizes plant growth independently of seasonal variations in controlled environment agriculture (CEA) systems [1]. In these systems, light-emitting diodes (LEDs) supply the specific light spectra required for plant growth and development [2]. This allows the establishment of ideal environmental conditions, resulting in rapid development, higher yields, and outstanding produce quality while lowering the demand for natural resources and mitigating the impacts of climate change [3,4]. Nevertheless, the expensive initial setup and operational expenses, precise control required over all the environmental cues, and the requirement for specialized expertise and technology have limited the widespread use of this technology for crop production [5].

Providing a proper lighting environment for crops is of immense importance in CEA. Several light attributes, including quality, intensity, and duration, influence the growth and development of crops under CEA conditions. The design and implementation of proper light quality in CEA has attracted attention nowadays. Light quality, including spectral composition and wavelength ratio, strongly affects plant growth, physiology, and development. For instance, red (R) light, with a wavelength between 600 and 700 nm [6], increases growth, photosynthetic capacity, and the production of secondary metabolites that are essential for flavor and food quality [7,8]. Despite some adverse reports related to its sole application, R light enhances plant photosynthesis by increasing the production of photosynthetic pigments, which in turn boosts CO_2_ uptake and leads to an increase in biomass [7]. Far-red (FR) light, with a wavelength between 700 and 800 nm, is widely acknowledged as an essential factor affecting plant growth physiology. FR light enhances vegetative growth and stimulates the reproductive development of horticultural crops [6,9]. FR light can either promote or suppress flowering, depending on its presence in the photoperiod [10]. It also contributes to the shade avoidance response. It influences the length of stems and the morphology of plants, resulting in an overall increase in total biomass and growth rate of plants [11]. Plants adjust their growth to compete for light in crowded environments with limited light intensity for normal photosynthesis and plant growth. Phytochromes are R and FR light photoreceptors that regulate plant development through their ability to switch between active (Pfr) and inactive (Pr) forms. This switching mechanism influences seed germination, shade avoidance, flowering, and biomass allocation [12]. The ratio of R to FR light influences the Phytochrome Photo-Stationary State (PSS), affecting the balance between the inactive and active forms of phytochromes (Pr and Pfr forms) [13]. This balance is essential, as it determines how plants respond to their light environment, making these light spectra vital for plant development and adaptation in CEA.

Night interruption (NI) is a horticulture method employed to change the plants’ photoperiod by supplying light during the night [14]. This approach subjects plants to night-time light exposure for a specific period, which affects their circadian rhythms and photoperiodic responses [15]. NI can manage energy consumption by changing certain parts of the lighting period to the dark period when demand and expenses are reduced, thus avoiding peak rates while maintaining optimal growth [16]. This method has been reported as a beneficial approach in CEA systems for maximizing plant development and yield [17,18].

Basil (*Ocimum basilicum* L.) is a member of the Lamiaceae family, which includes more than 150 species of aromatic plants. It is native to tropical regions of Asia and Africa [19]. Italian basil is well known for its distinct taste and essential oils. Italian basil cultivars exhibit variations in their physical traits and essential oil composition compared to other sweet basil varieties. They have larger leaves, with high levels of specific metabolites [20,21]. The production of Italian basil under artificial light in CEA has attracted attention. While there were prior studies involving light manipulation in other crops, the present study on night interruption with R and FR light usage in basil and its particular influence on phytochemical production presents new knowledge for research in CEA setups. The findings enable the further understanding of how light spectrum manipulation at night could optimize growth and quality in basil and further improve CEA practices at the commercial level. Therefore, the present study was designed to investigate the impact of NI using R and FR light spectra on the photosynthetic efficiency, biomass production, partitioning, and phytochemical production of Italian basil (*Ocimum basilicum* L.). Using NI with R and FR light spectra is expected to enhance plant growth and phytochemical production in Italian basil under CEA conditions.

## 2. Materials and Methods

### 2.1. Plant Material and Growth Conditions

The experiment was conducted at the Controlled Environment Agriculture Center (CEAC), Faculty of Agricultural Technology (Aburaihan), College of Agriculture and Natural Resources, University of Tehran, Iran. The seeds of Italian basil were initially planted in 50-cell trays filled with coir pith and perlite at a ratio of 1:2. The seeds were cultivated in a controlled environment under white light, at 25 ± 2 °C and relative humidity between 60 ± 10%. This ensured an optimal setting for germination and initial growth. Once the seeds had successfully germinated and the seedlings had four leaves, they were transplanted into pots with 15 cm diameters and 10 cm height. The pots were filled with a substrate-containing coir pith and perlite at a ratio of 1:1. This ratio promotes the growth of roots and provides stability for the plants. Subsequently, the plants were irrigated with Hoagland solution [22] by adjusting the pH to 5.7, and the electrical conductivity of the nutrient solution was 1.8 ds m^−1^.

### 2.2. Treatments

A mixture of red and blue light spectra was used at a ratio of 70% red light to 30% blue light (R-B), with an intensity of 200 µmol m^−2^ s^−1^ [23], provided by LED light modules (Parcham Company, Pakdasht, Tehran, Iran). The light period was 12 h (6 a.m. to 6 p.m.). This combination of light was used to grow all the plants during the light period. The experimental treatments included different NI lighting spectra, including NI with R light spectrum (NI-R), NI with FR light spectrum (NI-FR), NI with a combination of R and FR light spectra in a 1:1 ratio (NI-R-FR), and a control treatment without NI (C). During the dark period (6 p.m. to 6 a.m.), the NI was applied for two hours, from 12 a.m. to 2 a.m. The intensity of the R and FR light was 100 and 30 µmol m^−2^ s^−1^, respectively. The intensities for the R-FR were kept at 100 µmol m^−2^ s^−1^ by adjusting the height of the light modules from the plant canopy.

The treatments were continued for 4.5 weeks after applying. Finally, the collected samples were sent to the laboratory for additional analysis.

The daily light integral of 8.64 mol m^−2^ d^−1^ was maintained across all treatments, ensuring consistency in the amount of light each plant received apart from NI light treatments in the experiment. Therefore, 0.72 mol m^−2^ d^−1^ of DLI was added to the NI-R and NI-R-FR. The added DLI for NI-FR was 0.22 mol m^−2^ d^−1^. Light intensities and spectra were carefully monitored using a Sekonic light meter (Sekonic C-7000, Tokyo, Japan). The spectral composition of different light treatments is presented in Figure 1. PSSs for R-B, NI-R, NI-FR, and NI-R-FR were 0.87, 0.89, 0.14, and 0.87, respectively. The PSS was calculated based on the method described by Sager et al. [24]. The treatments were applied for 4.5 weeks before the collected samples were sent to the laboratory for further analysis. The growth chambers maintained an average temperature of 25  ±  2 °C, a CO_2_ concentration of 400 ± 50 ppm, and a relative humidity of 50  ±  5%.

### 2.3. Phytochemical Measurements

#### 2.3.1. Extraction

Young adult-developed leaves were picked from the plants. Liquid nitrogen was poured over them, and the samples were powdered using a mortar and pestle. The powdered samples were used for the extraction process. To measure total phenols and flavonoids, 300 mg of plant sample powder was mixed with 3 mL of 80% methanol (1:10 ratio). This mixture was placed in an ultrasonic device at 40 °C for 20 min [25]. Subsequently, the samples were centrifuged at 3000× *g* for 15 min, and the supernatant was separated as the extract for evaluating the compounds mentioned above.

150 mg of plant sample powder was mixed with 2 mL of ethanol acidified with 1% hydrochloric acid to prepare the extract for measuring anthocyanin content. This mixture was then placed in a shaker incubator at 4 °C for 24 h. The samples were centrifuged at 5500× *g* for 5 min, and the supernatant was used as the extract.

#### 2.3.2. Measurement of Phenols, Flavonoids, and Anthocyanins

The Folin-Ciocalteu reagent was used to assess the total phenol content by recording the absorbance of the samples at a wavelength of 730 nm. A calibration curve was drawn using gallic acid concentrations of 0, 100, 200, 300, 400, and 500 mg mL^−1^ [26]. The optical absorbance of flavonoids was measured at a wavelength of 415 nm to evaluate the flavonoid content. Different concentrations of quercetin were used to plot the standard curve [27]. To measure the anthocyanins content, the absorbance of the supernatant obtained from the extract in Section 2.3.1 was read at 530 nm and 657 nm [28]. The anthocyanin content was calculated using the following formula:Relative Anthocyanin Content=Absorbance at 530 nm−(Absorbance at 657 nm×0.25)

### 2.4. Measurement of Photosynthetic Pigments

To determine the amounts of chlorophyll a, chlorophyll b, and carotenoids, 150 mg of leaf powder was mixed with 2.5 mL of 80% acetone. The mixture was then centrifuged at 8000 rpm for 10 min. The supernatant was collected, and its absorbance was measured at wavelengths of 646 nm, 663 nm, and 470 nm. The concentrations of chlorophyll a, chlorophyll b, and carotenoids were calculated using the formulas provided by Lichtenthaler (1987) [29].

### 2.5. Post-Harvest Measurements

#### Leaf Desiccation Response

Leaf desiccation was used as an early postharvest response of excised leaf to postharvest conditions. Leaves were excised and re-cut under water. Plant leaves were first saturated for one hour at 21 °C and 35 µmol m^−2^ s^−1^ irradiance, provided by a white LED from Parcham Company, Pakdasht, Tehran, Iran, by placing them in a parafilm-closed container with degassed deionized water. Following saturation, the leaves were positioned upside down on balances in a controlled setting with 40 ± 3% relative humidity, 21 °C, 1.40 kPa VPD, and 35 µmol m^−2^ s^−1^ irradiance. The weight loss of the leaves was monitored gravimetrically for two hours using an HR200 scale with 0.0001 g accuracy. ImageJ v8 (National Institutes of Health, Bethesda, MD, USA) software determined the leaf area in scanned leaves. The transpiration rate was measured by a method described previously [30]. The transpiration rate was calculated using the following formula:$$TR=\frac{\Delta DW}{\Delta t}\times \frac{1}{M(H_{2}O)}\times 1000\times LA$$
where ΔDW is the difference in dry weight during desiccation, Δt is the time difference during measurement, $M(H_{2}O)$ is the molecular weight of water (18 g mol^−1^), and LA is the leaf area.

### 2.6. Biomass Partitioning

Plants were harvested, and the roots, stems, and leaves were separated. The plant components were dried in an oven at 75 °C for 72 h. The dry weights of leaves, roots, and stems were measured using an HR200 scale with 0.0001 g accuracy.

### 2.7. Photosynthetic Efficiency

Chlorophyll fluorescence was measured using a FluorPen PAR-FLORPEN FP 100-MAX device (Photon System Instrument, Brno, Czech Republic) using the OJIP protocol. The measurements were taken from young, fully developed leaves during week 4 of the vegetative growth stage, after the plants had been dark-adapted for at least 30 min. The dark adaptation was carried out in the middle of the light period. The Performance Index on Absorption Basis (PI_ABS_), the Dissipation per Reaction Center (DI_O_/RC), the minimum and maximum fluorescence yield of the dark-adapted state (F_o_ and F_m_), and the maximum quantum yield of Photosystem II (F_v_/F_m_) were calculated based on the setting of the device.

### 2.8. Statistical Analysis

This study used a completely randomized design, with four treatments and three replications (each replication included five plant samples). The data collected were analyzed using the SAS (Statistical Analysis System, version 9.4). Mean comparisons were performed and analyzed using Duncan’s multiple-range test. The correlation between traits was determined by calculating the Pearson correlation coefficients (r) using the R package (Version 4.3.3).

## 3. Results

### 3.1. Photosynthetic Pigments

Statistical analysis showed significant differences among the studied treatments for the photosynthetic pigments of Italian basil (*Ocimum basilicum* L.) exposed to different light conditions (Table 1).

The analysis of photosynthetic pigments revealed that the NI-R treatment significantly outperformed all other treatments across all four parameters measured (Figure 2). Specifically, chlorophyll content under NI-R reached 530 µg g FW^−1^, representing a 16.83% increase compared to the control. Similarly, chlorophyll b content was significantly enhanced, with 157.78 µg g FW^−1^ recorded, 20.6% higher than in the control. Carotenoid content also increased, reaching 101.23 µg g FW^−1^, reflecting a 10.9% rise over the control. The total photosynthetic pigment content under NI-R reached 788.4 µg g FW^−1^, a 16.7% increase relative to the control chlorophyll content.

### 3.2. Secondary Metabolites

Statistical analysis showed significant differences among studied treatments for the secondary metabolites of Italian basil exposed to different light conditions (Table 2). The highest anthocyanin content was recorded for NI-R, with 31.02 µg g FW^−1^ (Figure 3a), reflecting a 15.53% increase compared to the control, which recorded 26.85 µg g FW^−1^. Anthocyanin levels in NI-R were higher than in the other light conditions. For NI-R-FR, anthocyanin content was measured at 29.19 µg g FW^−1^, while NI-FR and C were relatively low, with 27.20 µg g FW^−1^ and 26.85 µg g FW^−1^, respectively. NI-R-FR showed the highest total phenol content, at 0.927 µg g FW^−1^, which is 109.73% higher than C, with a phenol content of 0.442 µg g FW^−1^ (Figure 3b). The flavonoid content in NI-FR was 1.45 µg g FW^−1^, which is 43.56% higher than in C (1.01 µg g FW^−1^). Flavonoid levels in NI-R and NI-R-FR were 1.36 µg g FW^−1^ and 1.33 µg g FW^−1^, respectively (Figure 3c).

### 3.3. Leaf Desiccation Response

To study the water loss property of the plants following excision from the plants, the transpiration rate of the excised leaf samples was tracked at the beginning and end of two hours of desiccation. The control samples had the highest transpiration rate, reaching a peak of around 3 mmol m^2^ s^−1^ at the beginning of the desiccation. On the other hand, the FR treatment (yellow line) had the lowest transpiration rate, approximately 1.5 mmol m^2^ s^−1^. Throughout the observation period, all treatments consistently decreased transpiration rates with different degrees of slope. The C treatment demonstrated the most significant decline with a slope of −0.0222, indicating the fastest rate of decrease. However, it consistently maintained the highest level, and ended with 0.4 mmol m^2^ s^−1^ of transpiration rate. In contrast, the FR light treatment had the most gradual decrease, with a slope of −0.0091, indicating a reduction of approximately 59.01% that was less steep than the C treatment. The R light treatment reached a transpiration rate of 0 by the end of the desiccation period (Figure 4).

### 3.4. Biomass Production and Partitioning

Statistical analysis showed significant differences among studied treatments for the dry weight of different organs of Italian basil (*Ocimum basilicum* L.) exposed to various light treatments (Table 3). The root, stem, and leaf dry weights demonstrated significant differences under the NI treatments compared to the control (Figure 5). The leaf dry weight showed the most difference, with NI-R-FR reaching 11.785 g, marking a 167.3% increase over the control, which recorded 4.41 g. Stem dry weight followed a similar trend, with NI-R-FR leading at 1.83 g, a 134.6% improvement compared to the control, which was closely matched by NI-R at 1.81 g. For root dry weight, NI-R-FR and NI-FR showed no significant differences, but they differed significantly from the control. NI-R-FR had the highest value at 1.5 g, representing a 92.3% increase compared to the control. The application of NI increased plant height (Figure 5d). The shortest plants were observed under control conditions, while the tallest plants were detected under NI application, including FR light (NI-R-FR and NI-FR).

In general, biomass production in plants exposed to NI-R-FR and NI-R showed more than two times the overall biomass of basil plants compared to the control (Figure 6a). In all treatments, the leaves had the highest biomass allocation. The NI-R-FR treatment had the highest leaf biomass proportion (78.49%), followed by NI-R (77.92%) and NI-FR (74.51%). The biomass contribution to the stem differed significantly among treatments. The NI-R treatment contributed the most stem biomass (13.53%), followed by NI-FR (12.56%) and NI-R-FR (11.52%). Additionally, NI-R contributed the smallest percentage of root biomass among all treatments. In contrast, the control had the highest root biomass proportion (13.13%), while NI-R-FR contributed the least (11.52%) (Figure 6b).

### 3.5. Photosynthetic Efficiency

Statistical analysis showed significant differences among studied treatments for some parameters obtained from the OJIP test of Italian basil (*Ocimum basilicum* L.) exposed to different light treatments (Table 4).

The OJIP test showed varying responses to the NI treatments. No significant difference was detected for F_v_/F_m_ in plants exposed to different NI light (Figure 7a). PI_ABS_ was highest in NI-R and NI-R-FR, measuring 1.7 and 1.75, respectively, with no significant difference. NI-FR, with a PI_ABS_ of 1.6, was not significantly different from NI-R or NI-R-FR, but it was significantly higher than the control with 1.45. For DI_O_/RC, the control and NI-FR had similar values with no significant differences. NI-R-FR (0.8) and NI-R (0.78) had significantly lower DI_O_/RC than the control, but NI-R-FR was not significantly different from the control or NI-FR. F*o* was highest in the control at approximately 15,000, while all NI treatments had lower values, around 13,000, indicating a significant decrease compared to the control. F_v_/F_m_ did not vary among treatments.

The results of the correlation analysis revealed significant relationships among various biochemical and physiological traits in Italian basil exposed to different light treatments (Table 5). There are high positive correlations between chlorophyll a (1) and chlorophyll b (2) (r = 0.99 **), as well as between chlorophyll b (2) and carotenoids (3) (r = 1.00 **). Anthocyanin content (4) shows strong negative correlations with DI_o_/RC (14) (r = 0.99 **). Total phenolic content (5) exhibits negative correlations with antioxidant activity (7) (r = −0.76**) and high positive correlations with leaf dry weight (8) (r = 0.96 *). Flavonoids (6) demonstrate significant positive correlations with leaf area (r = 0.98 *). Strong positive correlations exist among the following dry weight components: leaf dry weight (8), stem dry weight (9), root dry weight (10), and total plant dry weight (11). Notably, leaf dry and plant dry weights have a high correlation (r = 0.99 **). DI_o_/RC (14) shows a strong negative correlation with anthocyanin (4) (r = −0.99 **), while PI_ABS_ (15) is positively correlated with total phenolic content (5), and plant dry weight (11) (r = 0.98 *–0.99 **). F_0_ demonstrates strong positive correlations with antioxidant activity (7) (r = 0.99 **) and significant negative correlations with stem dry weight (9), plant dry weight (11), and leaf area (12) (r = −0.94 * to −0.97 **).

## 4. Discussion

In the present study, photosynthetic pigment levels were increased by NI-R, while using FR for NI hurt the pigment levels (Figure 2). Chlorophyll a and b are vital pigments that play a critical role in photosynthesis and are effective in light energy absorption. The observed increase in chlorophyll a and b content under NI-R treatment can be attributed to the high efficiency of photosynthesis influenced by R light [31,32]. The R light is recognized as a highly effective light spectrum for enhancing the photosynthetic process because it is absorbed by chlorophyll molecules, leading to increased light energy absorption in an efficient manner [33,34]. It has been reported that R light stimulates the biosynthesis of protochlorophyllide, a precursor in chlorophyll synthesis, thereby increasing chlorophyll content. In contrast, the lower chlorophyll content under NI-FR and NI-R-FR may be due to the inhibitory effect of FR light, which disrupts the biosynthetic pathway of chlorophyll precursors [35]. FR light is less effective in photosynthesis and more effective in morphological modification, which leads to a phenomenon known as the shade avoidance response. Under these conditions, plants prioritize stem growth to escape the shade of other plants, often limiting chlorophyll production [36,37]. It has been shown that R and FR light applications as NIs differentially affect the chlorophyll content of different plant species [38]. Short-day and long-day plants reduce their chlorophyll content when exposed to FR light for four hours during the night (dark) period. Conversely, R light in the context of NI increases the chlorophyll contents compared to FR light-exposed plants [38], which aligns with the findings reported in the present study (Figure 2).

Carotenoids are essential for photoprotection and light harvesting in photosynthesis. The highest carotenoid content was also detected under the NI-R treatment. This response is likely attributed to the role of R light in enhancing carotenoid biosynthesis pathways. Carotenoids act as accessory pigments, absorbing light energy and protecting chlorophyll from oxidative damage [39,40]. The lower carotenoid content under NI-FR may be due to reduced energy absorption and a decreased need for light protection, as FR has less energy than lower wavelengths. FR light significantly impacts the electron density in the photosynthetic reaction centers, particularly affecting Photosystem II (PSII) and Photosystem I (PSI), as it does not provide sufficient energy to effectively drive the water-splitting reaction, leading to fewer electrons entering the electron transport chain and a decrease in photosynthetic efficiency. Although PSI can absorb FR light more effectively, it still suffers from an electron deficiency due to the insufficient electron flow from PSII, resulting in the accumulation of oxidized electron carriers and further reducing the electron density in PSI reaction centers. This imbalance can increase the production of reactive oxygen species (ROS), which can damage cellular components and, if FR light predominates for an extended period, may reduce growth and productivity [41]. On the other hand, it has been shown that FR light reduces the electron density in reaction centers, thereby decreasing photoinhibition and potentially reducing the need for protective systems [42]. However, this should indicate that the intensity of FR light in the present study was much lower than the R light in the NI treatments. Having a higher light intensity for FR light is not considered rational since it would result in the extreme elongation of the plants, which results in lodging (unpublished data). The high positive correlations between chlorophylls and carotenoids (Table 5) indicate that effective light treatment enhances pigment synthesis, which is crucial for photosynthesis and photoprotection.

A balanced light environment can produce secondary metabolites, including antioxidants, at a standardized level, which increases with increased stress [43,44]. The present study’s findings indicate that plants under NI treatments did not perform better under extended light exposure as the result of two more hours of light. Since Italian basil is a facultative long-day plant, extending the light through NI increased photosynthetic efficiency and reduced short-light duration, thereby increasing antioxidant properties [45].

Anthocyanins are considered screens during light exposure. They play a role in light protection and the coloration of plant tissues. The highest anthocyanin content under the NI-R treatment may be explained by the activation of the anthocyanin biosynthesis pathway under R light. It has been reported that exposure to R light promotes the expression of genes involved in anthocyanin synthesis, such as chalcone synthase and dihydroflavonol-4-reductase [46,47]. Furthermore, R light also increases the synthesis of phenylalanine ammonia-lyase (PAL), a key enzyme in the phenylpropanoid pathway, leading to increased synthesis of phenolic compounds [45]. Phenolic compounds play a role in the plant’s defense system and its response to stress, and their production can be influenced by light-quality [48,49]. The results showing the highest flavonoid content under the NI-FR may be due to the plant’s response to FR light, which can stimulate flavonoid synthesis as part of the shade avoidance response [40,50]. The lower flavonoid content under the control treatment may be due to a balanced light environment without the additional influence of R or FR light. In a balanced situation, stimulating flavonoid biosynthesis is unnecessary for plant metabolite modification. A negative correlation exists between total phenolic content and antioxidant activity, while a positive correlation exists between total phenolic content and leaf dry weight (Table 5), indicating that phenolics may enhance structural growth without necessarily boosting antioxidant levels. The significant positive correlation between flavonoids and leaf area suggests that larger leaves can enhance flavonoid synthesis, contributing to plant health.

The biomass partitioning observed under various NI treatments highlights the critical role of R and FR light in regulating plant growth, mainly through their influence on the Phytochrome Steady State (PSS). The NI-R-FR treatment resulted in the highest total biomass and the most uniform distribution across plant organs [51]. This outcome can be attributed to the PSS, which reflects the balance between active (Pfr) and inactive (Pr) phytochrome forms. NI affects this balance by regulating the dynamic interconversion between Pr and Pfr. RL drives the conversion of Pr to Pfr, resulting in a higher P_fr_/P_Total_ ratio, while FRL induces the reverse process, shifting the PSS towards a lower P_Fr_/P_Total_ ratio [52]. The modulation of the phytochrome system during NI is crucial for certain physiological responses, such as shade avoidance. The increased leaf biomass observed in the NI-FR matches shade avoidance responses, as FRL induces elongation growth in low Pfr conditions [12]. The FR light exposure shifts the PSS towards a lower P_fr_/P_Total_ ratio, inducing shade avoidance responses that promote stem elongation and leaf surface expansion, resulting from the higher leaf biomass in this treatment [53]. R light, which increases the P_Fr_/P_Total_ ratio, enhances photosynthetic efficiency by stimulating cryptochromes and promoting CO_2_ uptake, leading to increased stem and leaf growth and root development by producing growth hormones like auxins [6,54]. The interaction between R light and FR light also influenced electron transport within photosynthetic reaction centers, with R light enhancing and FR light reducing efficiency, contributing to the observed growth patterns [55]. These findings underscore the importance of PSS and light spectra in shaping biomass partitioning, with the NI-R-FR treatment effectively optimizing plant growth and resource allocation [56,57]. However, it is worth noting that increased biomass production resulting from NI application can result from adding more light to the lighting environment. It has been reported that an extended photoperiod is more important than light intensity for biomass gain and phytochemical levels for basil production in controlled environments [58]. However, in the study by Eghbal et al. [58], an hourly increase in the duration of light from 12 to 18 h resulted in less than a 10% increase in biomass. Therefore, in our study, the increase in biomass mainly results from the NI application, and is not due to the extended duration. Future experiments are still needed to elucidate the role of NI in biomass and phytochemical production in basil plants. The same lights should be added to the growing light treatment for the daylight period with the NI light applications. Furthermore, they reported no negative impacts of FR light on the growth of basil plants. In their study, the FR light considerably decreased the leaf area, and the promotive effects of FR light on growth were mainly due to stem elongation. Therefore, in our study, the improved growth due to FR light is a result of its application as the NI.

It has been shown that R and FR light applications as the NI differentially affect the growth and flowering of different plant species [38]. Plants reduce their leaf area when exposed to FR for four hours during the night (dark) period. Conversely, R light used in NI increases the leaf area in different plant species [38], which aligns with the findings reported in the present study. Strong correlations among dry weight components of the present study (Table 5) reflect the interconnected nature of biomass accumulation, particularly the critical role of leaf dry weight in total plant growth.

The analysis of photosynthetic efficiency showed that the NI treatments significantly influenced various aspects of photosynthesis in basil plants. The reduction in F_o_ across NI treatments suggests enhanced photoprotective mechanisms or increased primary energy absorption efficiency through having more open reaction centers [57,59,60]. The consistent F_v_/F_m_ values between treatments indicate stable maximum quantum efficiency of PSII, suggesting that the overall photochemical conversion potential remains unaffected by R and FR light [61]. The lower DI_O_/RC observed in the NI-R and NI-R-FR treatments, compared to the control and NI-FR, indicates reduced energy loss per reaction center under R light conditions, highlighting more efficient energy management within the photosystems [62]. The strong negative correlation between anthocyanin content and DI_O_/RC (Table 5) suggests that increased anthocyanins, through providing a screen for extra light entrance, reduce energy dissipation from the photosynthetic apparatus. The higher PI_ABS_ in these treatments shows the enhanced performance of PSII under R light [63]. The positive correlation of PI_ABS_ with total phenolic content and plant dry weight (Table 5) indicates that effective light capture is associated with increased biomass. F0’s positive correlation with antioxidant activity and negative correlations with biomass suggest that higher antioxidant levels may be linked to stress responses that inhibit growth.

Exposure to NI-R light decreased the transpiration rate following two hours of desiccation. In comparison, the control showed the highest transpiration rate during two hours of leaf desiccation (Figure 4). Stomata are the primary path for leaf water loss. It has been reported that stomatal size decreases, leading to increased stomatal closing ability due to R light exposure [55]. It has been confirmed that a higher surface area to volume ratio in smaller stomata facilitates stomatal reactions [55]. Generating small stomata and elevated responsiveness to desiccation helps leaves keep internal water and decreases their vulnerability to water deficit and desiccation [55].

## 5. Conclusions

This study demonstrated that NI treatments significantly affected Italian basil’s phytochemical content and post-harvest water loss (*Ocimum basilicum* L.). The NI-R treatment significantly increased carotenoids, chlorophyll a, and chlorophyll b, ultimately increasing photosynthetic efficiency. This treatment also increased the anthocyanin content, showing the role of NI-R light in substantial biochemical processes. The NI-FR treatment was less effective in promoting chlorophyll content, but did stimulate flavonoid production. Regarding post-harvest quality, the desiccation response varied significantly among the NI treatments. The control had the highest transpiration rate, indicative of a higher rate of water loss. In contrast, the NI-FR treatment showed the least transpiration, indicating excellent moisture retention ability and a longer post-harvest life. Results showed that using correct NI treatment strategies with R and FR lights enhanced Italian basil’s photosynthetic efficiency and phytochemical production in CEA. From the results, it can be concluded that red and far-red light can improve plant growth and phytochemical yield. However, more research should determine the ideal NI duration, light intensity, and cost-effectiveness under commercial production conditions. Generalizing this research to other crops can improve CEA practices’ sustainability and commercial applicability.

## Figures and Tables

**Figure 1 plants-13-03145-f001:**
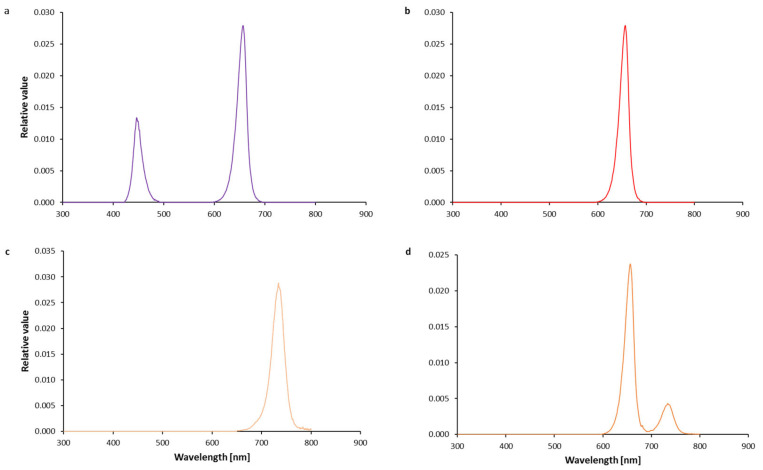
Spectral composition of light used as a control without NI [(**a**); red (peak wavelength at 660 nm) and blue (peak wavelength at 455 nm)], which was used for growing plants during the light period or as night interruption using different light spectra, including red light ((**b**); peak wavelength at 660 nm), far-red light ((**c**); peak wavelength at 630 nm), and a combination of both (**d**).

**Figure 2 plants-13-03145-f002:**
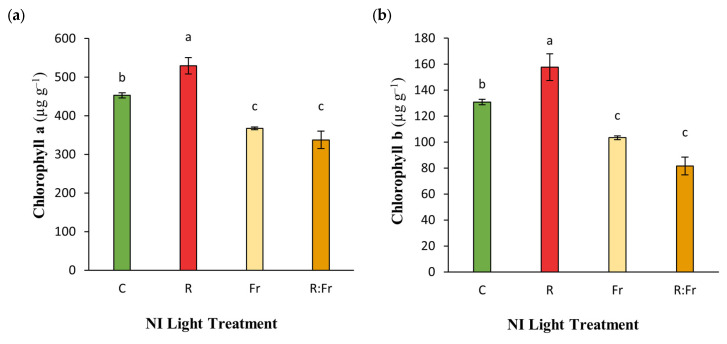
(**a**) Concentration of chlorophyll a; (**b**) chlorophyll b; (**c**) carotenoids; and (**d**) total photosynthetic pigments (chlorophyll a, b, and carotenoids) of Italian basil plants exposed to night interruption (NI) using different light spectra, including red light (R), far-red light (FR), a combination of both (R:Fr), and a control without NI (C). During the dark period, the NI treatments were used for 2 h. The intensity of the R and FR light was 100 and 30 µmol m^−2^ s^−1^, respectively. The intensities for the RFR treatment were kept at 100 µmol m^−2^ s^−1^ by adjusting the height of the light modules from the plant canopy. Columns are the mean value of three replications (each replication included five plants). Means with the same letters within the groups are not significantly different (*p* < 0.05).

**Figure 3 plants-13-03145-f003:**
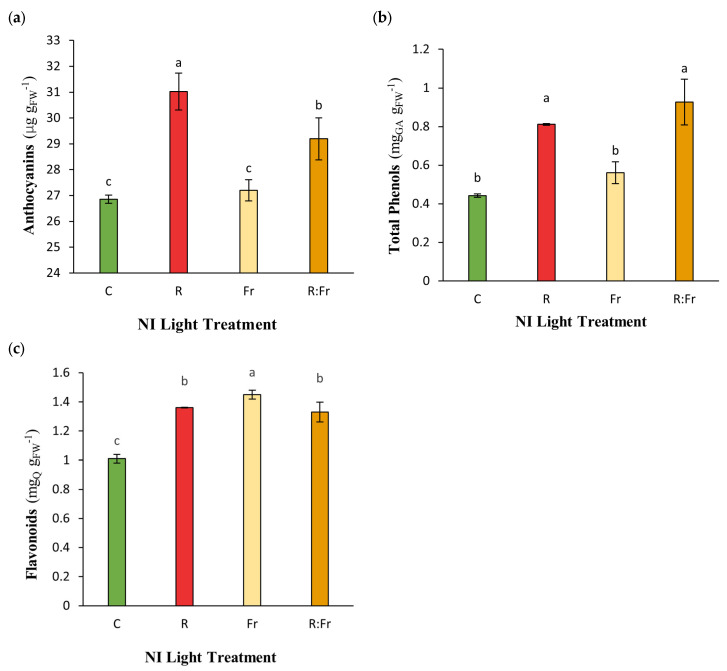
(**a**) Contents of anthocyanins; (**b**) total phenols; and (**c**) flavonoid of Italian basil plants exposed to night interruption (NI) using different light spectra. See Figure 1 and Figure 2 legends for details about light intensities and treatments.

**Figure 4 plants-13-03145-f004:**
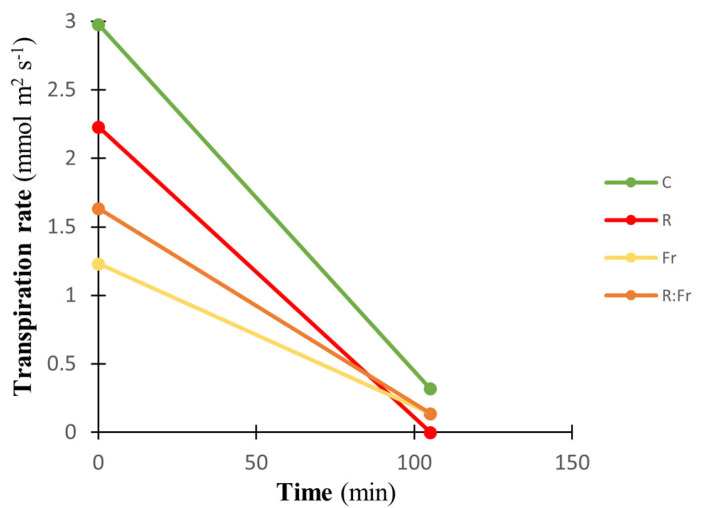
Transpiration rate during two hours of leaf desiccation of Italian basil plants exposed to night interruption (NI) using different light spectra. See Figure 1 and Figure 2 legends for details about light intensities and treatments. The leaves from different treatments were positioned upside down on balances in a controlled setting with 40 ± 3% relative humidity, 21 °C, 1.40 kPa VPD, and 35 µmol m^−2^ s^−1^ irradiance.

**Figure 5 plants-13-03145-f005:**
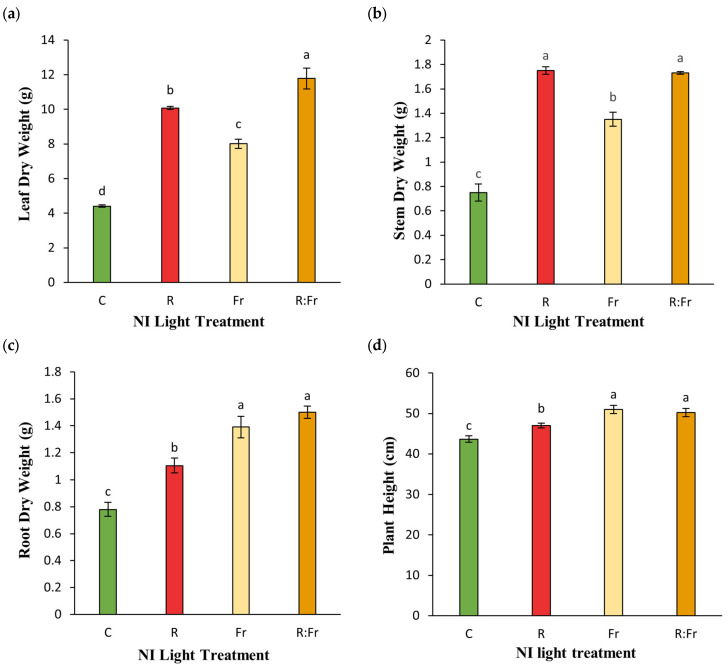
(**a**) Leaf dry weight; (**b**) stem dry weight; (**c**) root dry weight; and (**d**) plant height of Italian basil plants exposed to night interruption (NI) using different light spectra including. See Figure 1 and Figure 2 legends for details about light intensities and treatments. Columns are the mean value of three replications (each replication included five plants). Means with the same letters within the groups are not significantly different (*p* < 0.05).

**Figure 6 plants-13-03145-f006:**
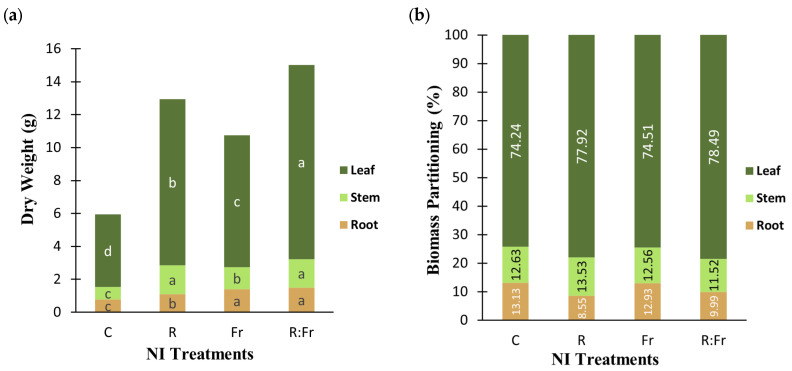
(**a**) Biomass production and (**b**) partitioning of different organs of Italian basil plants exposed to night interruption (NI) using different light spectra. See Figure 1 and Figure 2 legends for details about light intensities and treatments. Columns are the aggregate value of biomass (**a**) or percentage of biomass allocation to each organ (**b**) of plants exposed to different spectra of NI.

**Figure 7 plants-13-03145-f007:**
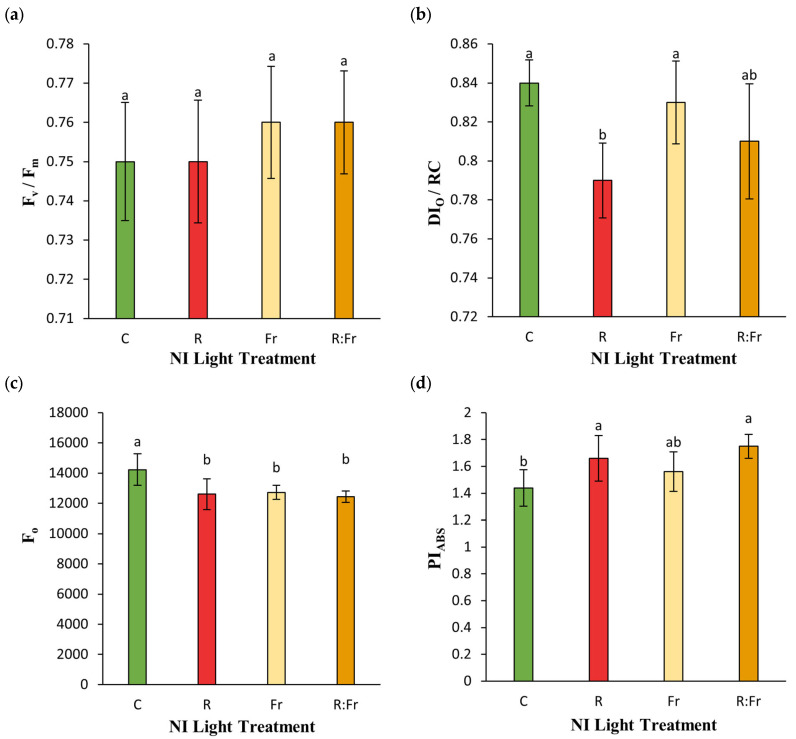
(**a**) Maximum quantum yield of photosystem II (F_v_/F_m_); (**b**) energy dissipation per reaction center (DI_O_/RC); (**c**) minimum fluorescence (F_o_); and (**d**) performance index per absorbed light (PI_ABS_) of Italian basil plants exposed to night interruption (NI) using different light spectra. See Figure 1 and Figure 2 legends for details about light intensities and treatments. Columns are the mean value of three replications (each replication included five plants). Means with the same letters within the groups are not significantly different (*p* < 0.05).

**Table 1 plants-13-03145-t001:** Statistical analysis results of the photosynthetic pigments of the Italian basil (*Ocimum basilicum* L.) exposed to different light treatments.

Sources of Variation	Degrees of Freedom	Mean Square			
		Chlorophyll a	Chlorophyll b	Carotenoid	Total Photosynthetic Pigments
Light Treatment	3	22,580.1 **	3287.17 **	404.76 **	50,380.21 **
Experimental Error	8	254.15	39.46	2.93	387.14

** means significant difference at 1% level.

**Table 2 plants-13-03145-t002:** Statistical analysis results of the secondary metabolites of the Italian basil (*Ocimum basilicum* L.) exposed to different light treatments.

Sources of Variation	Degrees of Freedom	Mean Square		
		Anthocyanin	Total Phenol	Flavenoids
Light Treatment	3	11.21 **	0.14 **	0.11 **
Experimental Error	8	0.34	0.004	0.001

** means significant difference at 1% level.

**Table 3 plants-13-03145-t003:** Statistical analysis results of leaf dry weight, stem dry weight, root dry weight of the Italian basil (*Ocimum basilicum* L.) exposed to different light treatments.

Sources of Variation	Degrees of Freedom	Mean Square			
		Leaf Dry Weight	Stem Dry Weight	Root Dry Weight	Plant Height
Light Treatment	3	15.73 **	0.311 **	730.08 **	266 *
Experimental Error	8	0.56	0.01	2.86	4.99

* and ** mean significant difference at 5% and 1% levels, respectively.

**Table 4 plants-13-03145-t004:** Statistical analysis results for some of the parameters obtained from the OJIP test of the Italian basil (*Ocimum basilicum* L.) exposed to different light treatments.

Sources of Variation	Degrees of Freedom	Mean Square			
		F_v_/F_m_	D_IO_/RC	PI_ABS_	F_o_
Light Treatment	3	91 × 10^−6 ns^	0.0026 *	0.089 *	3,455,077.98 *
Experimental Error	8	21 × 10^−6^	0.00045	0.019	6,929,404.07

ns and * mean no significant difference and 1% significance level, respectively.

**Table 5 plants-13-03145-t005:** Correlation matrix of biochemical and physiological traits in Italian basil exposed to different light treatments.

Correlations																
	1	2	3	4	5	6	7	8	9	10	11	12	13	14	15	16
Chlorophyll a (1)	1															
Chlorophyll b (2)	0.99 **	1														
Carotenoids (3)	0.99 **	1.00 **	1													
Anthocyanin (4)	0.46	0.38	0.38	1												
Total phenolic content (5)	−0.13	−0.23	−0.23	0.81	1											
Flavonoids (6)	−0.27	−0.24	−0.24	0.41	0.52	1										
Antioxidant activity (7)	0.40	0.42	0.42	−0.52	−0.76	−0.93	1									
Leaf dry weight (8)	−0.25	−0.32	−0.32	0.73	0.96 *	0.71	−0.90	1								
Stem dry weight (9)	−0.06	−0.12	−0.12	0.83	0.92	0.79	−0.90	0.96 *	1							
Root dry weight (10)	−0.70	−0.71	−0.71	0.22	0.64	0.82	−0.93	0.80	0.72	1						
Plant dry weight (11)	−0.27	−0.34	−0.34	0.72	0.95 *	0.75	−0.92	0.99 **	0.97 *	0.82	1					
Leaf area (12)	−0.35	−0.35	−0.35	0.46	0.64	0.98 *	−0.98	0.81	0.85	0.90	0.84	1				
F_v_/F_m_ (13)	−0.92	−0.90	−0.90	−0.22	0.30	0.62	−0.69	0.48	0.36	0.90	0.51	0.68	1			
DIo/RC (14)	−0.40	−0.32	−0.32	−0.99 **	−0.83	−0.51	0.60	−0.78	−0.88	−0.31	−0.77	−0.55	0.13	1		
PI_ABS_ (15)	−0.26	−0.34	−0.34	0.74	0.98 *	0.63	−0.86	0.99	0.94	0.76	0.99 **	0.75	0.45	−0.78	1	
F_0_ (16)	0.27	0.30	0.30	−0.63	−0.81	−0.92	0.99 **	−0.93	−0.95 *	−0.87	−0.94 *	−0.97 *	−0.59	0.71	−0.89	1

*. Correlation is significant at the 0.05 level (2-tailed). **. Correlation is significant at the 0.01 level (2-tailed).

## Data Availability

Data are contained within the article.

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
