# Peer review of "Night Interruption with Red and Far-Red Light Optimizes the Phytochemical Composition, Enhances Photosynthetic Efficiency, and Increases Biomass Partitioning in Italian Basil"

_plants, 2024, doi:10.3390/plants13223145_

Round 1
Reviewer 1 Report
Comments and Suggestions for Authors
1) The light quantum number in the range of 400-700 nm or 300-800 nm should be maintained in each experimental area, that is, the received daily cumulative light amount should be consistent in the whole experimental design. Otherwise, it is difficult to say whether the beneficial effect of each measurement, especially biomass, in the experiment is due to the effect of using different light quality to interrupt the dark phase, or the effect of additional red or far-red light. An additional 2 h of red light of 100 μmol m-2 s-1 is not negligible for dry matter increase.
2)For dark phase interruption, the intensity of monochromatic red light and monochromatic far red light are not consistent. For the result analysis of the measurement index, this does not indicate whether different light quality is used in the dark period or the amount of each light quality is caused. In the discussion section, the reasons for the differences of indicators under the three treatments of NI-R, NI-FR and NI-R-FR are ignored, and the results of each treatment are only described, which lacks logic and depth of discussion.
3)The whole article only describes the effects of dark phase interruption with different light qualities on the accumulation and distribution of photosynthetic pigments, secondary metabolites and dry matter, without further exploring the connection and interaction between these indicators. For the discussion part, the influence of red light or far red light on the result of a certain index is described too much, and it is ignored that this is the influence of dark period interruption processing. The effects of red light, far-red light and combined light should be described under the background of dark period interruption. At the same time, the discussion section does not discuss the results of other dark period interruption tests, no supporting literature for the results of this paper, and no discussion of the differences between the results of other papers.
4)Whether there is a connection between the measurement indicators of the article is debatable. The discussion should be focused, and the metrics need to serve that focus. In addition, it was mentioned several times in the discussion section that far red light can induce shade avoidance responses and affect plant morphology, which may be the reason for the increase in dry matter. However, the article lacks data related to plant morphology. If possible, please supplement it.
2.1 In Plant material and growth conditions, environmental parameters (temperature, humidity, CO2 concentration, etc.) during plant cultivation, irrigation frequency, irrigation amount, EC, pH and other information of nutrient solution should be added.
2.2 In Treatments, information such as measurement methods and instruments of light intensity and LED spectrum, manufacturer of LED lamps and photoelectric characteristics should be added. For the used LED light quality, its spectral distribution information should be increased, such as wavelength range, central wavelength, etc.
Line 87:The location of the experiment should be given. Line 151: The manufacturer, model and other information of the instrument and software should be given. Line 153: Specify the type and manufacturer of the dry weighing instrument.
Line 103:Is "Red and FR light spectra in a 1:1 ratio" the ratio of the number of light quanta in the respective wavelength range of Red and FR?
Line 104:"The NI was used for two hours." How long after the dark period began, the NI was used.
Line 151:When calculating the "transpiration rate" of postharvest leaves, the necessary calculation formulas should be listed.
Line 187:The error line is missing in Fig. 1d.
Line 275:The significance analysis of data between treatments needs to be added in Fig. 5.
Author Response
Reviewer 1
Thanks for the recommendations and comments. We believe that your suggestion helped us to improve the manuscript. We have considered all your suggestions, and the revisions are track changed in the revised manuscript. A point-by-point response is provided in the following:
The light quantum number in the range of 400-700 nm or 300-800 nm should be maintained in each experimental area, that is, the received daily cumulative light amount should be consistent in the whole experimental design. Otherwise, it is difficult to say whether the beneficial effect of each measurement, especially biomass, in the experiment is due to the effect of using different light quality to interrupt the dark phase, or the effect of additional red or far-red light. An additional 2 h of red light of 100 μmol m-2 s-1 is not negligible for dry matter increase.
Response: Thank you for noticing this important issue. In the revised manuscript, we have provided further information about the daily light integral (DLI) during growth and by the NI applications.
This study was performed following another study by our team (https://doi.org/10.1186/s12870-024-05637-w), which showed that duration but not intensity of light is more important for basil production. However, as you indicated, 100 μmol m-2 s-1 is not negligible for biomass gaining. Considering your remark, we have added the following text to the discussion and made some suggestions for future studies as well:
"However, it is worth noting that increased biomass production owing to NI application can be a consequence of enhancing DLI. It has been reported that a longer photoperiod is more important than light intensity for biomass gaining and phytochemical level for basil production in controlled environments [62]. However, in the study of Eghbal et al. [62], a six-hour increase in the duration of light from 12 to 18 hours resulted in less than a 10% increase in biomass. Therefore, in the present study, the increase in biomass is mainly because of the NI application, not due to extended duration. Future experiments are still needed to elucidate the role of NI on biomass and phytochemical production in basil plants. Together with the NI light applications, the same lights should be added to the growing light treatment for the daylight period. Furthermore, Eghbal et al. [62] reported no negative impacts of FR on the growth of basil plants. In their study, the FR considerably decreased the leaf area, and the promotive effects of FR on growth were mainly due to stem elongation. Therefore, in our study, the improved growth due to FR is due to its application as the NI."
In response to your concern regarding the consistency of light intensities, we would like to emphasize that the daily cumulative light quantum within the 400-500 nm range for blue light (BL), the 600-700 nm range for red light (RL), and the 700-800 nm range for far-red light (FR) was maintained for the growth of all treatments during day time, then for the NI different treatments were applied. Please note that we have considered any light condition as a treatment. The provided wavelength graphs are now available in Figure 1 of the revised manuscript.
For dark phase interruption, the intensity of monochromatic red light and monochromatic far-red light are not consistent. For the result analysis of the measurement index, this does not indicate whether different light quality is used in the dark period or the amount of each light quality is caused. In the discussion section, the reasons for the differences of indicators under the three treatments of NI-R, NI-FR and NI-R-FR are ignored, and the results of each treatment are only described, which lacks logic and depth of discussion.
Response: Thanks for noticing this. The layout of the present study was designed based on our previous study. Higher light intensity for the FR seems irrational because it would substantially affect leaf area production based on our last report (https://doi.org/10.1186/s12870-024-05637-w). We have added some reasoning for why different light intensities for R and FR light are used (the modifications can be seen in the text in the first response). Also, the issue related to light quality is now addressed in different parts of the discussion.
The whole article only describes the effects of dark phase interruption with different light qualities on the accumulation and distribution of photosynthetic pigments, secondary metabolites and dry matter, without further exploring the connection and interaction between these indicators. For the discussion part, the influence of red light or far red light on the result of a certain index is described too much, and it is ignored that this is the influence of dark period interruption processing. The effects of red light, far-red light and combined light should be described under the background of dark period interruption. At the same time, the discussion section does not discuss the results of other dark period interruption tests, no supporting literature for the results of this paper, and no discussion of the differences between the results of other papers.
Response: To address your criticism points, we have tested the correlation analysis among different indicators, added it as Table 5, and discussed this information in the discussion section. We also elaborate on the discussion section regarding the NI using different light qualities and compare the results with previous publications. These modifications can be seen in addition to the materials and methods (statistical analysis part), results (Table 1 and its description), and discussion (different additions to the discussion) of the revised manuscript.
Whether there is a connection between the measurement indicators of the article is debatable. The discussion should be focused, and the metrics need to serve that focus. In addition, it was mentioned several times in the discussion section that far red light can induce shade avoidance responses and affect plant morphology, which may be the reason for the increase in dry matter. However, the article lacks data related to plant morphology. If possible, please supplement it.
Response: As described in the previous response, we tested the correlation analysis among different indicators and added it to Table 1. For the morphological data, since the FR mainly influenced the height of the plants, we added the plant height graph to Figure 4.
In Plant material and growth conditions, environmental parameters (temperature, humidity, CO2 concentration, etc.) during plant cultivation, irrigation frequency, irrigation amount, EC, pH and other information of nutrient solution should be added.
Response: As requested, we have included the necessary environmental parameters in the revised manuscript's Materials and Methods section.
2.2 In Treatments, information such as measurement methods and instruments of light intensity and LED spectrum, manufacturer of LED lamps and photoelectric characteristics should be added. For the used LED light quality, its spectral distribution information should be increased, such as wavelength range, central wavelength, etc.
Response: We have provided accurate data on the light intensities and spectral distributions as measured using the Sekonic C-7000 light meter (Sekonic, Japan), ensuring that the overall setup meets the requirements for our experimental design.
Additionally, the spectral distribution information, including the wavelength ranges, is already documented in Figure 1 of the revised manuscript.
Line 87:The location of the experiment should be given. Line 151: The manufacturer, model and other information of the instrument and software should be given. Line 153: Specify the type and manufacturer of the dry weighing instrument.
Response: The revised manuscript has been updated with the location of the experiment, the manufacturer and model of the dry weighing instrument (AND HR200), the software used for leaf area measurements (ImageJ v8, NIH, USA), and other related information.
Line 103:Is "Red and FR light spectra in a 1:1 ratio" the ratio of the number of light quanta in the respective wavelength range of Red and FR?
Response: The 1:1 ratio refers to the number of LEDs used for Red and Far-Red light. The reason for using an equal number of LEDs is based on the practical setup of the experiment. Red and Far-Red light have different effects on plant growth, with Far-Red light typically requiring lower intensity to trigger biological responses. Therefore, while the number of LEDs was equal, the light intensities were not, with Red light at 100 μmol m⁻² s⁻¹ and Far-Red at 30 μmol m⁻² s⁻¹. This difference was intentional and designed to meet the plants' specific needs, and the overall light output was carefully measured using a Sekonic C-7000 light meter. We have provided the spectral distribution graph (Figure 1 of the revised manuscript) to the revised manuscript.
Line 104:"The NI was used for two hours." How long after the dark period began, the NI was used.
Response: The NI was applied for two hours, from 12 a.m. to 2 a.m., and it was exactly started in the middle of the 12-hour dark period (6 p.m. to 6 a.m.). This timing was chosen to ensure that the interruption occurred during the study's most relevant part of the dark period. To address your concern, we have clarified this in the manuscript (part 2.2).
Line 151:When calculating the "transpiration rate" of postharvest leaves, the necessary calculation formulas should be listed.
Response: The formula for calculating the transpiration rate has been added to the revised manuscript.
Line 187:The error line is missing in Fig. 1d.
Response: The error line has now been added.
Line 275:The significance analysis of data between treatments needs to be added in Fig. 5.
Response: The significance analysis for Figure 5(a) has now been added.

Reviewer 2 Report
Comments and Suggestions for Authors
Comments to the MS of S. Fallah et al.
Night Interruption with Red and Far-Red Light Optimizes the 1 Phytochemical Composition, Enhances Photosynthetic Efficiency, and Increases Biomass Partitioning of Italian Basil
The MS is devoted to an important topic- improving production quality and increasing plant biomass. However the MS contains a number of lacks.
Main problem is the incomplete description of phytochromes role in night interruption by RL and FRL. Which processes are under phytochromes control? Please, describe that in more detail.
In addition, One should theoretical novelty in this study
Minor comments:
1. I would advise to authors to include in the Abstract some specific numbers. For example, how great was the increase in pigments?
One should show the charaсteristics of RED light and far-red light (maximum in emission spectrum and half-width of the emission bandwidth.
Fig. 1. It is written on Y axis photosynthetic pigments. I think that authors mean Chl (a+b) content.
Fig. 2 (a,b) .It is better to written: “ Y axis anthocyanins and total phenols.. In caption please write “contents “ instead of concentrations.
Fig. 2. I think that content of anthocyanins is to high. Likely it should be as mkg /g (FW). Please. Test this point.
Lines 103-104. It is written: During the dark period, the NI was used for two hours. When was used NI during night period at what time?
Lines 104 and 105. Why were RL and FRL used at 100 and 30 μmol (photons) m-2 s-1, respectively and why RL/FRL ratio was 1:1?
Please, write equally, for example, R-FR instead of R-FR, RFR, RFr etc.
Authors write in all legends to figures: “During the dark period, the NI treatments were used for 2 h. The intensity of the R and FR light was 100 and 30 μmol m-2 s-1, respectively. The intensities for the R:Fr treatment were kept at 100 μmol m-2 s-1 through adjusting the height of the light modules from the plant canopy”. This is extra information. It is better to write that in Fig. 1 and add “and in other figure legends.
Lines 105 and 106. It is written: The intensities for the RFR were kept at 100 μmol m-2 s-1. It is better to write: It is better to write: The intensities for the R-FR were kept at 100 μmol m-2 s-1.
Line 112. It is written: To measure total phenol, flavonoid content, and antioxidant capacity, However, I did find a figure or table with such data. Please, test that again.
Please, correct English. For examples: Line 364. Anthocyanins considered as screen during light exposure. Probably one should write: Anthocyanins are considered. Line 401. It is written: The lower DIO/RC observed in the NI-R and NI-RFR treatments, compared to the control and NI-FR, indicate reduced energy loss per reaction center under R light conditions. Please, correct that. One should write: “The lower DIO/RC observed in the NI-R and NI-RFR treatments, compared to the control and NI-FR, indicates reduced energy loss per reaction center under R light conditions.
Conclusion. The MS presents many data and compares them with a lot of papers. It is OK. However, it needs some work and can be considered again after improving
Comments on the Quality of English LanguageQuality of English is OK. There are only minor lacks.
C
Author Response
Reviewer 2
Thanks for the recommendations and comments. We believe that your suggestion helped us to improve the manuscript. We have considered all your suggestions, and the revisions are track changed in the revised manuscript. A point-by-point response is provided in the following:
Comments and Suggestions for Authors
Comments to the MS of S. Fallah et al.
Night Interruption with Red and Far-Red Light Optimizes the 1 Phytochemical Composition, Enhances Photosynthetic Efficiency, and Increases Biomass Partitioning of Italian Basil
The MS is devoted to an important topic- improving production quality and increasing plant biomass. However, the MS contains a number of lacks.
Main problem is the incomplete description of phytochromes role in night interruption by RL and FRL. Which processes are under phytochromes control? Please, describe that in more detail.
In addition, One should theoretical novelty in this study
Response: Thank you for your comment. As requested, we have added a more detailed explanation of the role of phytochromes in the night interruption treatments and their influence on biomass partitioning and shade avoidance response. The modifications can be seen in the introduction (lines 62-67) and in the discussion:
Some other information related to the phytochrome photostationary state and other extra information has also been added to the materials and methods used in the revised manuscript.
This is a novel work, considering the approach of RL and FRL night interruption on growth, photosynthetic efficiency, and phytochemical production in Italian basil. While there were prior studies involving light manipulation in other crops, the present study of night interruption with RL and FRL usage in basil and their particular influence on phytochemical production presents new knowledge for controlled environment agriculture (CEA) setups. The findings enable further understanding of how light spectrum manipulation at night could optimize growth and quality in basil and further improve CEA practices at the commercial level. We modified the last paragraph of the introduction section to elaborate more on the novel aspect of the present study.
Minor comments:
- I would advise to authors to include in the Abstract some specific numbers. For example, how great was the increase in pigments?
One should show the characteristics of RED light and far-red light (maximum in emission spectrum and half-width of the emission bandwidth.
Response: In the abstract, we now include specific numbers (percentage of changes) for some studied traits.
Figure 1 provides the characteristics of Red and Far-Red light, including the spectral distribution. The necessary details have also been added to the article. The peak wavelength is now added to the abstract and to the materials and methods.
Fig. 1. It is written on Y axis photosynthetic pigments. I think that authors mean Chl (a+b) content.
Response: The Y-axis label " Total Photosynthetic Pigments" includes chlorophyll a, b, and carotenoids, as all these are photosynthetic pigments. Carotenoids are essential for photoprotection and light harvesting in the photosynthetic process. We have added more information to the Figure legend for clarification.
Fig. 2 (a,b) .It is better to written: “ Y axis anthocyanins and total phenols.. In caption please write “contents “ instead of concentrations.
Response: Since the units’ indications are different, we kept the y-axis separately for the anthocyanins and total phenols. The caption has been updated as suggested.
Fig. 2. I think that content of anthocyanins is to high. Likely it should be as mkg /g (FW). Please. Test this point.
Response: The unit was intended to be in micrograms (µg g FW-1). The typo has been corrected.
Lines 103-104. It is written: During the dark period, the NI was used for two hours. When was used NI during night period at what time?
Response: The timing of the night interruption has been clarified in the manuscript. The NI was applied for two hours, from 12 a.m. to 2 a.m., which started in the middle of the 12-hour dark period (6 p.m. to 6 a.m.). This timing was chosen to ensure that the interruption occurred during the study's most relevant part of the dark period. To address your concern, we have clarified this in the manuscript (part 2.2).
Lines 104 and 105. Why were RL and FRL used at 100 and 30 μmol (photons) m-2 s-1,
respectively and why RL/FRL ratio was 1:1?
Please, write equally, for example, R-FR instead of R-FR, RFR, RFr etc.
Response: The intensities of 100 μmol m⁻² s⁻¹ for RL and 30 μmol m⁻² s⁻¹ for FRL were chosen to ensure that the experimental setup accurately reflects the intended effects of the light spectra on the plants, but without the assumption that the intensities alone are directly effective in influencing plant responses. Therefore, the ratio of RL to FRL is not about the effective intensities but rather the number of LEDs used to maintain the desired light quality. The spectral distribution is now added as Figure 1 to the revised manuscript for more clarification.
The layout of the present study was designed based on our previous study. Having a higher light intensity for the FR did not seem rational because, based on our last report (https://doi.org/10.1186/s12870-024-05637-w), it would strongly affect leaf area production. We have added some reasoning for why different light intensities for R and FR light were used.
Consistent indications for NI light treatments were considered whole through the manuscript
Authors write in all legends to figures: “During the dark period, the NI treatments were used for 2 h. The intensity of the R and FR light was 100 and 30 μmol m-2 s-1, respectively. The intensities for the R:Fr treatment were kept at 100 μmol m-2 s-1 through adjusting the height of the light modules from the plant canopy”. This is extra information. It is better to write that in Fig. 1 and add “and in other figure legends.
Response: We have simplified the figure legends as recommended. The detailed information about light intensities and treatments is now only in Figure 1 and 2. We added "See Fig. 1 and 2 legends for details about light intensities and treatments” for all other figures.
Lines 105 and 106. It is written: The intensities for the RFR were kept at 100 μmol m-2 s-1. It is better to write: It is better to write: The intensities for the R-FR were kept at 100 μmol m-2 s-1.
Response: The RFR has been changed to R-FR throughout the manuscript.
Line 112. It is written: To measure total phenol, flavonoid content, and antioxidant capacity, However, I did find a figure or table with such data. Please, test that again.
Response: Thank you for your observation. The revised manuscript does not mention "antioxidant capacity."
Please, correct English. For examples: Line 364. Anthocyanins considered as screen during light exposure. Probably one should write: Anthocyanins are considered. Line 401. It is written: The lower DIO/RC observed in the NI-R and NI-RFR treatments, compared to the control and NI-FR, indicate reduced energy loss per reaction center under R light conditions. Please, correct that. One should write: “The lower DIO/RC observed in the NI-R and NI-RFR treatments, compared to the control and NI-FR, indicates reduced energy loss per reaction center under R light conditions.
Response: Thank you for the suggestions. The suggestions are fully considered and applied in the revised manuscript.

Reviewer 3 Report
Comments and Suggestions for Authors
To the authors
The submitted paper is very interesting because it shows that a short time of light irradiation has a significant effect. However, the minimum information necessary for readers to check and evaluate the data, such as the method, is not included. This is a major problem for an academic paper.
I hope that you will take the following points into consideration and prepare a revised manuscript.
1. The term “Night interruption (NI)”, which is also in the title, should be avoided. Currently, at least in the agricultural field, the term “Night break (NB)” is used to refer to short time of light irradiation near the center of the dark period. In the past, plant physiologists used the term “Light interruption” to refer to the process of irradiating light in the middle of the night. However, although it does mean blocking by light, the time of irradiation is unclear. Therefore, “Night break” came to be used to mean dividing the dark period into two by irradiating light near the center of the dark period. The NI used by the authors is an ambiguous expression and should be avoided.
2. Regarding the definition of light, the far-red is 700-750 nm, but this is commonly 700-800 nm.
3. The light source for the plant cultivation conditions in the experimental method is not specified. It is not specified whether it is a fluorescent light or an LED, and the irradiation time is not specified. It is a well-known fact that when conducting research to examine the effects of light quality, the subsequent treatment results change significantly depending on the seedling cultivation conditions. Similarly, the light source is not specified in the experiment to look at leaf desiccation.
4. The method for measuring the PPFD of light is not specified. This also needs to be specified in order to ensure reliability, as the measurement accuracy varies greatly depending on the measuring device. In addition, FR cannot be measured with a normal photon sensor (which measures PPFD). The time of NI treatment is also not specified, which is a serious problem.
5. The method for harvesting and powdering the samples is not specified. There is no description of how the fresh weight was measured for each part 4.5 weeks after the start of treatment, and then the dry weight was measured and powdered, but these are not described.
6. The centrifugal force is shown in “rpm”, but of course if the rotation radius is different, the centrifugal force will be different even at the same rotation speed (rpm), so it should be expressed in “x g”.
7. The time when the photosynthetic efficiency was measured is not specified. Naturally, this is a parameter that changes during the growth stage, so describing is essential.
8. Duncan's statistical method was used for multiple testing, but currently, it has been pointed out that this method has problems and should be avoided. The method used by Turkey’s or others should be used.
9. The units in the results shown in Fig. 2 (mg)do not match those described in the text (ug). Also, although part of the polyphenols (total phenols) are flavonoids, and part of the flavonoids are anthocyanins, the values in the figure are completely opposite.
Comments on the Quality of English LanguageGood
Author Response
Reviewer 3
Thanks for the recommendations and comments. We believe that your suggestion helped us to improve the manuscript. We have considered all your suggestions, and the revisions are track changed in the revised manuscript. A point-by-point response is provided in the following:
The submitted paper is very interesting because it shows that a short time of light irradiation has a significant effect. However, the minimum information necessary for readers to check and evaluate the data, such as the method, is not included. This is a major problem for an academic paper.
I hope that you will take the following points into consideration and prepare a revised manuscript.
- The term “Night interruption (NI)”, which is also in the title, should be avoided. Currently, at least in the agricultural field, the term “Night break (NB)” is used to refer to short time of light irradiation near the center of the dark period. In the past, plant physiologists used the term “Light interruption” to refer to the process of irradiating light in the middle of the night. However, although it does mean blocking by light, the time of irradiation is unclear. Therefore, “Night break” came to be used to mean dividing the dark period into two by irradiating light near the center of the dark period. The NI used by the authors is an ambiguous expression and should be avoided.
Response: Thank you for your valuable suggestion concerning the use of terms. After going through the difference that several academic articles made between "Night Break" and "Night Interruption," we considered your point seriously. Although the term "Night Break (NB)" generally means a short pulse of light in the middle of the dark period, "Night Interruption (NI)" is considered herein as a more general term that is often used to describe light exposure during the night that might vary in both duration and timing. This can be seen in Li et al. 2021, Park et al. 2016, and Park & Jeong 2020 and in the famous book of Kozai: Kozai, T. Smart Plant Factory, The Next Generation Indoor Vertical Farms. Springer 2018. We also recognize concern over the ambiguity of this period and have taken care to specify the duration of night light exposure in our study so as not to create ambiguity. Since our experiment involves continuous light exposure through the night, we believe Night Interruption describes our methodology more precisely and, as such, has retained the term in the manuscript. To make it all clear, in the materials and methods of the revised manuscript, we included further information about the timing of the application of NI to clarify and remove ambiguities.
- Regarding the definition of light, the far-red is 700-750 nm, but this is commonly 700-800 nm.
Response: Thank you for bringing this to our attention. We have checked the spectral distribution of FR and figured out that there are some wavelength detections till 760 nm. Therefore, the range of the far-red light is revised. We have also included a graph (Figure 1 of the revised manuscript) demonstrating the spectral distribution based on wavelength for more clarity. Further information regarding the measurements and their precision was added to the manuscript to maintain consistency and transparency in our methodology.
- The light source for the plant cultivation conditions in the experimental method is not specified. It is not specified whether it is a fluorescent light or an LED, and the irradiation time is not specified. It is a well-known fact that when conducting research to examine the effects of light quality, the subsequent treatment results change significantly depending on the seedling cultivation conditions. Similarly, the light source is not specified in the experiment to look at leaf desiccation.
Response: Thanks for noticing this. We added the type of light source in the experimental methods section. Moreover, we have included the irradiation time and other related information about the light treatment in the materials and methods. In addition, we have added a light source needed in desiccation.
- The method for measuring the PPFD of light is not specified. This also needs to be specified in order to ensure reliability, as the measurement accuracy varies greatly depending on the measuring device. In addition, FR cannot be measured with a normal photon sensor (which measures PPFD). The time of NI treatment is also not specified, which is a serious problem.
Response: Thank you for your comment. We have now specified the method for measuring PPFD. The light intensities, including far-red light, were measured using a Sekonic C-7000 light meter, which can accurately measure both visible and far-red light (from 380 to 800 nm). Additionally, the timing of the Night Interruption (NI) treatment, from 12 a.m. to 2 a.m., has been added to the manuscript for clarity.
- The method for harvesting and powdering the samples is not specified. There is no description of how the fresh weight was measured for each part 4.5 weeks after the start of treatment, and then the dry weight was measured and powdered, but these are not described.
Response: We have now added the requested details. Both fresh and dry weights were measured using the AND HR200 scale, and the revised manuscript clarifies the process for harvesting, drying, and powdering the samples.
- The centrifugal force is shown in “rpm”, but of course if the rotation radius is different, the centrifugal force will be different even at the same rotation speed (rpm), so it should be expressed in “x g”.
Response: The unit has been revised based on the "x g".
- The time when the photosynthetic efficiency was measured is not specified. Naturally, this is a parameter that changes during the growth stage, so describing is essential.
Response: We agree. The timing of the photosynthetic efficiency measurements has been added to the materials and method section of the revised manuscript.
- Duncan's statistical method was used for multiple testing, but currently, it has been pointed out that this method has problems and should be avoided. The method used by Turkey’s or others should be used.
Response: Thank you for suggesting using Duncan's method for multiple comparisons. Indeed, this method is widely utilized in agricultural studies and other fields and is recognized as an effective tool for comparing means. Duncan's method has specific advantages, including handling unequal sample sizes and providing interpretable results under various conditions. We understand some researchers may prefer alternative methods such as Tukey's HSD. However, ultimately, the choice of the appropriate method depends on the nature of the data and the research objectives.
- The units in the results shown in Fig. 2 (mg) do not match those described in the text (ug). Also, although part of the polyphenols (total phenols) are flavonoids, and part of the flavonoids are anthocyanins, the values in the figure are completely opposite.
Response: The confusion was caused by a typo in the figure. The unit for anthocyanins is µg gFW-1, which has been corrected. For total phenols, the unit is mgGA gFW-1 (gallic acid equivalents), as explained in the related citation, and for flavonoids, the unit is mgQ gFW-1(quercetin equivalents).

Round 2
Reviewer 1 Report
Comments and Suggestions for Authors
none
Comments on the Quality of English LanguageMinor editing of English language required.